# PURE: Prototypical Mutual Prompting Enhancement for Zero-shot Text-attributed Graph Learning

## Abstract

This paper studies the problem of zero-shot text-attributed graph learning, which aims to generate high-quality node representations in unseen text-attributed graphs. Recent approaches usually utilize large language models (LLMs) instead of graph neural networks (GNNs) to extract semantics due to their strong generalization ability, which could neglect the intrinsic geometric structure. Towards this end, we propose a novel approach named Prototypical Mutual Prompting Enhancement (PURE) for zero-shot text-attributed graph learning. The core of our PURE is to generate high-quality prompts using prototypical learning to combine the advantages of both language models and graph models. In particular, we first utilize dual graph pre-training from both instance and informativeness perspectives to generate a generalizable GNN. Then, we incorporate the frozen language and graph models into a mutual prompt learning framework. On the one hand, we extract node tokens with geometric relationships using the graph model, which will be sent to multiple prototypical projections to enhance the understanding of the language model. On the other hand, we extract graph information and task descriptions using the language model, which serves as instruction for the graph models. Extensive experiments on both node classification and link predictions validate the effectiveness of PURE compared to competing baselines.

## 1 Introduction

Graph serves as a versatile data structure for effectively capturing intricate relationships and dependencies. A notable variation is the Text-Attributed Graph (TAG), where textual information, such as node or edge descriptions, is incorporated into the graph to enhance its data representation. This integration makes TAGs particularly valuable in various domains, including social network analysis (Backstrom & Leskovec, 2011) and recommender systems (Wang et al., 2020). Graph models, especially Graph Neural Networks (GNNs), have achieved remarkable performance and become a *de facto* approach for graph-based machine learning. Despite great success, these graph models are usually trained or fine-tuned for a particular dataset or task, struggling to maintain consistent performance when applied to new datasets or tasks (Ju et al., 2023; Li et al., 2024).

Fortunately, the emergence of Large Language Models (LLMs) has significantly advanced the zero-shot capabilities of machine learning models. By leveraging vast amounts of encoded pre-existing knowledge, LLMs can effectively generalize to new datasets or tasks, making them highly adaptable across various fields. For instance, in the natural language processing (NLP) field, models like GPT-4 (Achiam et al., 2023) and Llama (Touvron et al., 2023) unify all the tasks as a generative paradigm, allowing them to handle tasks they have never seen before. In the computer vision (CV) field, models such as CLIP (Radford et al., 2021) employ a retrieval-based approach, mapping images and textual descriptions into a shared embedding space to enable zero-shot recognition of new images by comparing their similarity to textual labels. However, since LLMs are designed for sequential text modeling, directly applying them to graph-related tasks presents new challenges, particularly in encoding the structural information of graphs.

In recent years, leveraging the strength of LLMs for graph models has sparked growing interest. LLM as Enhancer (Yu et al., 2023; Chen et al., 2024c; Liu et al., 2024) leverages language models instead

of traditional shallow embedding methods like Bag of Words (BoW) to enrich the graph feature space. These approaches have shown promising performance since their effectiveness in capturing semantic nuances, but they are still constrained by their reliance on GNNs for final predictions. LLM as Aligner (Wen & Fang, 2023) further maps both graph and corresponding text modalities into a shared embedding space, focusing on transferring pre-trained models within the same graph. In contrast, LLM as Predictor (Guo et al., 2023; Fatemi et al., 2023) directly translates graph data into plain texts suitable for LLMs and uses them for specific predictions, leveraging zero-shot capabilities of LLM for graph tasks.

Despite the promising performance of these methods, formalizing a framework for zero-shot graph learning remains challenging since two questions are required to tackle: ❶ *How to leverage the GNN to generate the transferable representation of the intrinsic graph structure?* Graphs inherently contain complex structural dependencies that traditional LLMs may not capture effectively. The GNN model needs to be fine-tuned to generate strong expressive embeddings that can be generalized across tasks and domains. ❷ *How to integrate the transferable representation into LLM that works effectively for zero-shot graph learning?* Unlike structured graph models, LLMs operate on sequential text-based inputs. This presents a challenge in aligning graph-generated embeddings, which capture the structural dependencies of the graph, with the LLM model to ensure that the graph's information is effectively interpreted and utilized by the LLM for zero-shot learning tasks.

Towards this end, in this paper, we propose a novel approach named **P**rototypical **MU**tual P**R**ompting **E**nhancement (termed PURE), which combines the advantages of both GNNs and LLMs to generate high-quality prompts for zero-shot text-attributed graph learning. Specifically, we first perform the dual graph pre-training, which considers two perspectives. The instance view focuses on learning node representations based on immediate neighbors to capture the structural relationships in the graph. The informativeness view emphasizes identifying and leveraging the parts most relevant to the LLM token embeddings for alignment between the two models. Then, we integrate the frozen graph and language models into a mutual prompt learning framework. On the one hand, the graph model extracts node tokens with geometric relationships, which are then passed through prototypical projections to transform the graph into a more comprehensible format to enhance the LLM model. On the other hand, the LLM processes graph-related information and task descriptions, providing high-level instructions as the prompt to enhance the graph model. This mutual prompting not only improves the interaction between models but also boosts the overall zero-shot graph learning performance.

The contribution of the paper can be summarized as follows: (1) *New Connection.* We pioneer a new perspective to utilize prompt learning to combine the advantages of both language models and graph models for zero-shot text-attributed graph learning. (2) *Novel Methodology.* Our PURE not only leverages graph models to extract geometric relationships for language model prompting, but also generates text-based prompting using prototypical projections for graph model enhancement. (3) *Extensive Experiments.* Extensive experiments on both node classification and link predictions validate the superiority of our proposed PURE. Our code is available at https://anonymous. 4open.science/r/PURE.

## 2 RELATED WORK

### 2.1 PROMPT LEARNING FOR GNNS

Prompt learning for GNNs has evolved from simple feature augmentation to increasingly sophisticated designs. Early works introduced learnable prompt tokens to node features for pre-training alignment (Fang et al., 2022; Shirkavand & Huang, 2023), later extended with multiple prompt tokens for greater flexibility (Fang et al., 2024). Subsequent approaches diversified the prompt space: view-specific prompts (Gong et al., 2023), subgraph-based or task-specific prompts (Sun et al., 2023; Huang et al., 2024), and edge-level prompt tuning such as EdgePrompt, which learns prompt vectors for edges to enhance message passing (Fu et al., 2025). In parallel, benchmark efforts like ProG standardize evaluation of diverse prompting methods (Zi et al., 2024), while theoretical analyses explain prompting's ability to approximate graph transformations (Wang et al., 2024b). Specialized frameworks target particular settings, including heterogeneous graphs (HetGPT (Ma et al., 2024)), dual-task prompting during pre-training (ULTRA-DP (Chen et al., 2023)), and self-adaptive prompts leveraging pre-training components (Self-Pro (Gong et al., 2024)). However, GNNs' limited parameter capacity compared to LLMs still restricts their ability to fully exploit prompt learning. Our

work addresses this by constructing GNN prompts through interaction with LLMs, enabling GNNs to benefit from the capacity of LLMs.

## 2.2 GRAPH ALIGNMENT WITH LLMS

Integrating LLMs with graph-structured data combines their generalization and relational reasoning abilities. A common approach converts graphs into textual representations for LLM input (Guo et al., 2023; Chen et al., 2024b; Liu et al., 2024), but often loses structural properties. Recent works (Tang et al., 2024a;b; Chai et al., 2023; Fatemi et al., 2023) instead use GNNs as structural encoders to align graph data with LLMs. Molecular graph–text integration follows a similar trend, with MolCA and InstructMol bridging molecular structures and natural language via contrastive and multi-task pretraining (Liu et al., 2023; Cao et al., 2023). Recently, other approaches have enhanced GNN–LLM synergy by injecting language semantics to improve structural representations and by incorporating structured knowledge directly into LLMs via GNNs (Li et al., 2025a;b). Beyond alignment, large models can also act as controllers for automated GNN design. LLM4GNAS (Gao et al., 2025) integrates an LLM into the Graph Neural Architecture Search process to automate feature engineering and hyperparameter optimization. Despite these advances, most methods rely on unidirectional alignment, limiting integration and joint optimization. We propose a mutual prompting framework enabling bidirectional exchange between GNNs and LLMs, enhancing alignment and generalization.

## 3 NOTATIONS & PROBLEM DEFINITION

**Notations.** Let a graph be denoted as $\mathcal{G} = (\mathcal{V}, \mathcal{E}, \boldsymbol{A}, \boldsymbol{X})$, where $\mathcal{V}$ is the node set with $N$ nodes and $\mathcal{E} \subseteq \mathcal{V} \times \mathcal{V}$ is the edge set. We use the adjacency matrix $\boldsymbol{A} \in \{0,1\}^{N \times N}$ to describe the structural information of the graph, where $\boldsymbol{A}_{uv} = 1$ if $(u,v) \in \mathcal{E}$, otherwise $\boldsymbol{A}_{uv} = 0$. The node feature matrix is given by $\boldsymbol{X} \in \mathbb{R}^{N \times F}$, where each row $\boldsymbol{x}_v \in \mathbb{R}^F$ corresponds to the $F$-dimensional vector containing attribute information of node $v$. For the node classification task, each node $v$ is assigned a label $y_v \in \mathcal{Y}$.

**Graph Neural Networks.** Graph Neural Networks (GNNs) have become a foundational framework for learning effective representations of graph-structured data. By employing a message-passing paradigm, GNNs iteratively update node representations by embedding both the graph topology and node features. Specifically, at the $l$-th layer, each node $v \in \mathcal{V}$ aggregates information from its neighbors $\mathcal{N}_v$ and combines it with its previous-layer embedding $\boldsymbol{h}_v^{(l-1)}$ for update:

$$\boldsymbol{z}_v^{(l)} = \mathcal{C}^{(l)}\Big(\boldsymbol{z}_v^{(l-1)}, \mathcal{A}^{(l)}\Big(\big\{\boldsymbol{z}_u^{(l-1)}\big\}_{u \in \mathcal{N}_v}\Big)\Big), \tag{1}$$

where $\mathcal{A}^{(l)}$ and $\mathcal{C}^{(l)}$ are the two functions that aggregate and combine embedding from the neighborhood. By iteratively stacking $L$ message-passing layers, the node representation can be $Z = \{\mathbf{z}_1^{(L)}, \ldots, \mathbf{z}_N^{(L)}\} \in \mathbb{R}^{N \times F_G}$, where $F_G$ denotes the representation dimension.

**Zero-Shot Graph Learning.** Recently, zero-shot learning has been developed in areas like image and text data, enabling models to generalize to new classes or tasks without relying on labeled data from the target domain. In this work, we aim to study zero-shot learning for graph data, with a particular emphasis on *cross-dataset* and *cross-task* scenarios. For *cross-dataset zero-shot learning*, we train a classification model on a fully labeled source graph $\mathcal{G}^s$ and test it on a completely different target graph $\mathcal{G}^t$, where $\mathcal{G}^s \cap \mathcal{G}^t = \emptyset$ and $\mathcal{Y}_s \cap \mathcal{Y}_t = \emptyset$. For *cross-task zero-shot learning*, we directly apply the model trained on the node classification task to the link prediction task without any fine-tuning.

## 4 THE PROPOSED PURE

### 4.1 FRAMEWORK OVERVIEW

The overview of our proposed zero-shot graph learning framework is illustrated in Figure 1. By pre-training the GNN model and aligning it with the LLM through mutual prompting, the `PURE` is capable of exhibiting substantial zero-shot learning abilities in both cross-dataset and cross-task scenarios. Our `PURE` framework consists of two phases. The GNN model is first pre-trained with the LLM's token embeddings from both instance and informativeness perspectives to capture graph structural

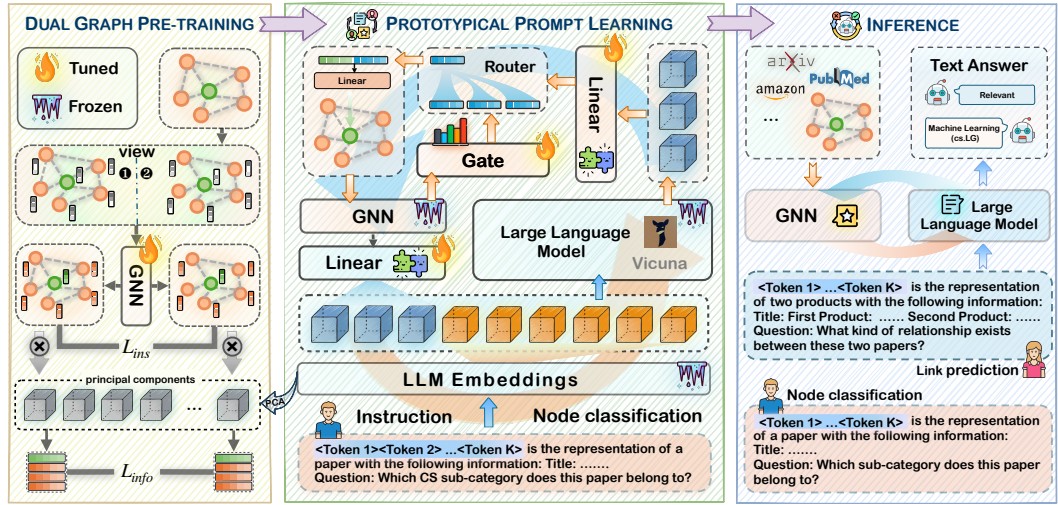

Figure 1: The overall framework of the proposed PURE. The framework consists of two phases: (1) Dual graph pre-training captures instance-level structural relationships and informativeness-aware semantic alignment with LLM token embeddings. (2) Mutual prompt learning enables iterative enhancement where GNN-derived tokens guide LLM understanding through prototypical projections, while LLM-generated instructions enhance GNN performance via specialized prompt experts.

relationships while identifying and leveraging the parts relevant to the LLM token embeddings for alignment between the two models (see Section 4.2). Then, we align the GNN with the LLM in an iterative mutual prompt learning manner to effectively transfer knowledge between the two models. On the one hand, we extract node tokens with geometric relationships and pass these tokens through prototypical projections, which transform the graph into a more comprehensible format to enhance the LLM model. On the other hand, the LLM processes graph-related information and task descriptions, generating high-level instructions as prompts to further enhance the graph model (see Section 4.3).

## 4.2 DUAL GRAPH PRE-TRAINING FOR GENERALIZABLE GNNs

In this part, we introduce a graph pre-training strategy to capture the transferable node representations suitable for alignment with LLMs. In general, there are several efforts proposed to construct self-supervised pretext tasks for pre-training GNNs, especially contrastive methods (Zhu et al., 2020), which offer broader applicability and overlapping task sub-spaces for better knowledge transfer.

**Instance-aware Pre-training.** Given the training graph, we adopt the Removing Edges (RE) and Masking Node Features (MF) strategies to generate two different graph views. For the RE strategy, we generate a random masking matrix $\widetilde{\boldsymbol{R}} \in \{0, 1\}^{N \times N}$, with each entry sampled from a Bernoulli distribution $\widetilde{\boldsymbol{R}}_{ij} \sim \mathcal{B}(1 - p_r)$ to mask edges with probability $p_r$, which can be computed as:

$$\widetilde{\boldsymbol{A}} = \boldsymbol{A} \circ \widetilde{\boldsymbol{R}}. \tag{2}$$

where $\circ$ denotes the Hadamard product. Similarly, for the MF strategy, we generate a random masking vector $\widetilde{\boldsymbol{m}} \in \mathbb{R}^F$ from another Bernoulli distribution $\widetilde{\boldsymbol{m}}_i \sim \mathcal{B}(1 - p_m)$ with probability $p_m$. The masked node feature $\widetilde{\boldsymbol{X}}$ can be:

$$\widetilde{\boldsymbol{X}} = [\boldsymbol{x}_1 \circ \widetilde{\boldsymbol{m}}; \cdots; \boldsymbol{x}_N \circ \widetilde{\boldsymbol{m}}]^\mathrm{T}. \tag{3}$$

The two views of the graph can be generated as $\widetilde{\mathcal{G}}_1 = (\widetilde{\boldsymbol{A}}_1, \widetilde{\boldsymbol{X}}_1)$ and $\widetilde{\mathcal{G}}_2 = (\widetilde{\boldsymbol{A}}_2, \widetilde{\boldsymbol{X}}_2)$. Then, we encode the two graph views to get the node embeddings, denoted as:

$$\boldsymbol{Z}^* = f_{\mathrm{GNN}}(\widetilde{\boldsymbol{A}}^*, \widetilde{\boldsymbol{X}}^*), * \in \{1, 2\}, \tag{4}$$

where $\boldsymbol{Z}^* = \{\boldsymbol{z}_1^*, \ldots, \boldsymbol{z}_N^*\} \in \mathbb{R}^{N \times F_G}$. We further employ a contrastive objective to distinguish the embeddings of the same node in two different views from those of other nodes:

$$\ell(\boldsymbol{z}_v^1, \boldsymbol{z}_v^2) = \log\left( \frac{\phi(\boldsymbol{z}_v^1, \boldsymbol{z}_v^2)}{\sum_{u=1}^{N} \phi(\boldsymbol{z}_v^1, \boldsymbol{z}_u^1) + \sum_{u \neq v} \phi(\boldsymbol{z}_v^1, \boldsymbol{z}_u^2)} \right), \tag{5}$$

where $\phi(\boldsymbol{z}_u, \boldsymbol{z}_v) = \exp(\boldsymbol{z}_u \cdot \boldsymbol{z}_v / \tau)$ with temperature parameter $\tau$. The objective from the instance perspective is:

$$\mathcal{L}_{ins} = \frac{1}{2N} \sum_{v=1}^{N} [\ell(\boldsymbol{z}_v^1, \boldsymbol{z}_v^2) + \ell(\boldsymbol{z}_v^2, \boldsymbol{z}_v^1)]. \tag{6}$$

**Informativeness-aware Pre-training.** Since a notable discrepancy exists between the node representations and the semantic space of LLMs, we introduce informativeness-aware contrastive learning with token embeddings to bridge this gap. Specifically, we employ principal component analysis (PCA) to extract the top $P$ principal components $\boldsymbol{C} \in \mathbb{R}^{P \times F_L}$ from the token embeddings of LLMs, where $F_L$ is the token embedding dimension. These components represent the directions that maximize variance in token embeddings and serve as coordinate axes for aligning node representations with the textual embedding space. We then map the node representations to the space as:

$$\boldsymbol{Z}^* = \boldsymbol{Z}^* \times \boldsymbol{C}^{\mathrm{T}}. \tag{7}$$

In practice, we set $F_G = F_L$ to facilitate mapping. And we break the independence between nodes to conduct the informativeness-aware contrastive learning:

$$\mathcal{L}_{info} = \frac{1}{F_L} \sum_{i=1}^{F_L} \frac{\phi(\boldsymbol{u}_i^1, \boldsymbol{u}_i^2)}{\sum_{j=1}^{F_L} [\phi(\boldsymbol{u}_i^1, \boldsymbol{u}_j^1) + \phi(\boldsymbol{u}_i^1, \boldsymbol{u}_j^2)]}, \tag{8}$$

where $(\boldsymbol{Z}^*)^{\mathrm{T}} = \{\boldsymbol{u}_1^*, \dots, \boldsymbol{u}_P^*\} \in \mathbb{R}^{P \times N}$. The final objective for token-aligned graph pre-training is:

$$\mathcal{L} = \frac{1}{2}(\mathcal{L}_{ins} + \mathcal{L}_{info}) \tag{9}$$

## 4.3 Prototypical Prompt Learning for Mutual Enhancement

The advent of LLMs has provided a new approach for graph learning. However, existing research (Huang et al., 2023) suggests that LLMs alone are insufficient for fully comprehending the graph data. Thus, given the pre-trained GNN, we aim to learn transferable prompts that enable effective mutual alignment between the GNN and LLM models.

**Geometric Prompting for Language Model.** To enable the LLM to capture graph data more effectively and enhance its performance in zero-shot graph learning tasks, we introduce a new graph-guided prompt tuning that includes specially designed instructions. Here, the instruction can be divided into two parts. We first provide the context to describe the graph information and then introduce the goal of the task. For the graph information, the encoded graph representations from the pre-trained GNN are utilized to construct the soft prompt. We further add the node's text attribute to enhance the LLMs' understanding. The graph information in the instructions can be presented as follows: $\langle\text{graph}\rangle$ *is/are the representation(s) of a paper/two papers/a paper set with the following information: Title: First paper:* $\{\text{title}_1\} \dots \backslash n$, where $\langle\text{graph}\rangle$ and $\{\text{title}_1\}$ denote the placeholders for both graph representation and text description inputs. Given the intricate nature of graphs and their diverse semantics, relying on a single prompt instruction may fail to cover the entire prompt space, thereby limiting the model's ability to capture the full spectrum of information on the targeted task. To address this, we introduce a set of GNN prototypes to characterize the entire prompt space, dividing it into several homogeneous regions, with each region being handled by a specialized prototype. Given the dual graph pre-training, the linear projector is sufficient to capture the mapping relationship:

$$\boldsymbol{H}_v = \{\boldsymbol{h}_v^1, \dots, \boldsymbol{h}_v^K\}, \quad \boldsymbol{h}_v^k = f_{\text{Linear}}^{G,k}(\boldsymbol{z}_v), \tag{10}$$

where $\boldsymbol{H}_v \in \mathbb{R}^{K \times F_L}$ with $K$ distinct space, $\boldsymbol{h}_v^k \in \mathbb{R}^{F_L}$ is the projected $k$-th node embedding for the LLM, $f_{\text{Linear}}^{G,k}(\cdot)$ denote the $k$-th linear project function. In this way, we replace $\langle\text{graph}\rangle$ with $K$ token embeddings as the soft prompt for the LLM, and the output token can be seen as diverse experts for the prompt of the GNN model. For the task descriptions, we directly add the question and alternative answers for the task to construct the instruction. Take the node classification task as an example. The instruction can be formulated as follows: *Which category does this paper belong to? Please directly choose the most likely answer from the following categories:* $\{\text{ans}\}$, where $\{\text{ans}\}$ here represents all the alternative answers and varies across datasets.

**Language Prompting for Graph Model.** Since the LLM encodes both graph information and task descriptions, its semantic richness can be utilized to guide the generation of graph prompts that emphasize class-specific representations. Specifically, let $\boldsymbol{h}_k \in \mathbb{R}^{F_L}$ represent the $k$-th output token corresponding to $\langle \text{graph} \rangle$ in the LLM. We apply an additional projector to map $\boldsymbol{h}_k$ to the $k$-th soft graph prompt $\boldsymbol{p}^k$, which can be defined as:

$$\boldsymbol{P} = \{\boldsymbol{p}^1, \ldots, \boldsymbol{p}^K\}, \quad \boldsymbol{p}^k = f_{\text{Linear}}^{L_1,k}(\boldsymbol{h}_k), \tag{11}$$

where $\boldsymbol{P} \in \mathbb{R}^{K \times F_L}$, $f_{\text{Linear}}^{L_1}(\cdot)$ denote the $k$-th linear projectors. The output tokens can be seen as diverse experts for the prompt of the GNN model. We then introduce a gating mechanism with a router model that decides how the input should be directed to the appropriate soft prompt, depending on the relevant semantic context. The prototypical weight of the $k$-th prompt can be:

$$w_k(\boldsymbol{z}_v) = \left[\text{Softmax}(f_{\text{Linear}}^R(\boldsymbol{z}_v) \circ (1 + \delta))\right]_k, \tag{12}$$

where $\delta \sim \mathcal{N}(0, 1)$ denotes the scaled Gaussian noise to encourage exploration of inputs over diverse prompts. The final soft prompt is utilized as the weighted combination of the prompt set, which is defined as follows:

$$\boldsymbol{x}'_v = [\boldsymbol{x}_v; \boldsymbol{w}^{\text{T}}\boldsymbol{P}], \boldsymbol{x}_{v,p} = f_{\text{Linear}}^{L_2}(\boldsymbol{x}'_v), \tag{13}$$

where $\boldsymbol{w} \in \mathbb{R}^K = \{w_1, \ldots, w_K\}$, $[\boldsymbol{x}; \boldsymbol{y}]$ here denotes the concatenation operation between $\boldsymbol{x}$ and $\boldsymbol{y}$. Note that through $K$ prompt experts, we can allow each expert to focus on and specialize in a specific region to enable the model's generalization. $\boldsymbol{x}_{v,p}$ denote the prompted node feature. Note that through $K$ prompt experts, we can allow each expert to focus on and specialize in a specific region to enable the model's generalization.

**Regularization to Mitigate Collapse.** To prevent a trivial solution where only one group of experts is consistently selected, we introduce two additional regularizations. For the geometric prompting in the LLM model, we enforce that the projected $K$ token embeddings are independent of each other by introducing a constraint of orthogonality between each token. The independent loss can be:

$$\mathcal{L}_{ind} = \frac{1}{N} \sum_{v \in \mathcal{V}} |\boldsymbol{H}_v \boldsymbol{H}_v^{\text{T}} - I|, \tag{14}$$

where $|\cdot|$ is the $L_1$ norm and $I$ denotes the identify matrix. For language prompting in the GNN model, the importance loss of each expert can be:

$$\text{Imp}(w)_k = \sum_{v \in \mathcal{V}} (w_k(\boldsymbol{z}_v)), \ \mathcal{L}_{imp} = \text{CV}(\text{Imp}(w))^2, \tag{15}$$

where $\text{CV}(\cdot)$ represents the coefficient of variation. Here the importance loss measures the variation of routing probabilities and enforces each expert to be similarly important.

### 4.4 Model Training and Evaluation

To facilitate the iterative mutual prompt tuning between the GNN and LLM models, we first utilize the embedding layer of Vicuna-7B-v1.5 (Zheng et al., 2023) to encode raw text as node features, followed by dual graph contrastive learning from both instance and informativeness perspectives to pre-train the GNN model. During each time of mutual alignment tuning, we freeze the GNN model, leverage the prompted node feature matrix $\boldsymbol{X}_p$, and train the corresponding linear projectors, along with a prompt router, using the dual graph pre-training loss (Equation 9) and the relevant regularization (Equation 14) for language prompting in the GNN model. Then, we in turn freeze the LLM model and directly train the linear projector on the downstream-specific task within the same dataset, as well as the corresponding regularization (Equation 15) for the geometric prompting in the LLM model. We denote the number of iterative mutual alignment tuning times as $I$. Finally, we evaluate the model performance on unseen datasets and tasks.

### 4.5 Theoretical Analysis of PURE

Here, we provide a theoretical analysis of `PURE`, demonstrating that the lack of a mixture of prompting expert strategies can limit the model's ability to capture task complexity, introducing bias in the promoted node feature matrix $\boldsymbol{X}_p$.

Table 1: **Cross-dataset zero-shot accuracy** on citation and e-commerce datasets (bold highlights the best result across all methods, while underline highlights the second-best results).

| Model | Pubmed | Cora | Children | History | Photo | Sports |
|---|---|---|---|---|---|---|
| MLP | $0.323 \pm 0.027$ | $0.021 \pm 0.006$ | $0.029 \pm 0.037$ | $0.080 \pm 0.041$ | $0.110 \pm 0.070$ | $0.042 \pm 0.021$ |
| *GNN as Predictor* | | | | | | |
| GCN | $0.288 \pm 0.092$ | $0.017 \pm 0.004$ | $0.030 \pm 0.018$ | $0.063 \pm 0.042$ | $0.103 \pm 0.047$ | $0.042 \pm 0.025$ |
| GraphSAGE | $0.316 \pm 0.058$ | $0.014 \pm 0.007$ | $0.008 \pm 0.007$ | $0.195 \pm 0.206$ | $0.056 \pm 0.055$ | $0.051 \pm 0.015$ |
| GAT | $0.343 \pm 0.064$ | $0.016 \pm 0.004$ | $0.086 \pm 0.084$ | $0.172 \pm 0.098$ | $0.050 \pm 0.027$ | $0.142 \pm 0.138$ |
| DGI | $0.329 \pm 0.103$ | $0.026 \pm 0.009$ | $0.082 \pm 0.035$ | $0.218 \pm 0.168$ | $0.224 \pm 0.127$ | $0.049 \pm 0.017$ |
| GKD | $0.399 \pm 0.033$ | $0.042 \pm 0.008$ | $0.202 \pm 0.064$ | $0.339 \pm 0.138$ | $0.166 \pm 0.086$ | $0.208 \pm 0.077$ |
| GLNN | $0.390 \pm 0.011$ | $0.031 \pm 0.006$ | $0.187 \pm 0.012$ | $0.283 \pm 0.102$ | $0.140 \pm 0.019$ | $0.317 \pm 0.048$ |
| NodeFormer | $0.308 \pm 0.093$ | $0.018 \pm 0.007$ | $0.048 \pm 0.022$ | $0.168 \pm 0.127$ | $0.073 \pm 0.015$ | $0.165 \pm 0.057$ |
| DIFFormer | $0.361 \pm 0.071$ | $0.029 \pm 0.014$ | $0.129 \pm 0.030$ | $0.275 \pm 0.171$ | $0.311 \pm 0.025$ | $0.306 \pm 0.131$ |
| OFA | $0.314 \pm 0.059$ | $0.130 \pm 0.019$ | $0.064 \pm 0.086$ | $0.052 \pm 0.049$ | $0.340 \pm 0.026$ | $0.101 \pm 0.071$ |
| *LLM as Predictor* | | | | | | |
| Vicuna-7B-v1.5 | $0.719 \pm 0.010$ | $0.156 \pm 0.001$ | $0.270 \pm 0.011$ | $0.363 \pm 0.001$ | $0.378 \pm 0.004$ | $0.370 \pm 0.001$ |
| Vicuna-7B-SPT | $0.768 \pm 0.036$ | $0.168 \pm 0.018$ | $0.227 \pm 0.015$ | $0.281 \pm 0.088$ | $0.350 \pm 0.061$ | $0.230 \pm 0.018$ |
| GraphGPT-std | $0.701$ | $0.126$ | - | - | - | - |
| GraphGPT-cot | $0.521$ | **$0.181$** | - | - | - | - |
| LLaGA | $0.793 \pm 0.036$ | $0.168 \pm 0.032$ | $0.199 \pm 0.007$ | $0.146 \pm 0.067$ | $0.276 \pm 0.069$ | $0.352 \pm 0.033$ |
| TEA-GLM | $0.839 \pm 0.013$ | $0.164 \pm 0.010$ | $0.271 \pm 0.010$ | $0.528 \pm 0.058$ | $0.497 \pm 0.027$ | $0.404 \pm 0.010$ |
| PURE | **$0.845 \pm 0.010$** | $0.173 \pm 0.008$ | **$0.275 \pm 0.015$** | **$0.594 \pm 0.050$** | **$0.520 \pm 0.015$** | **$0.436 \pm 0.024$** |

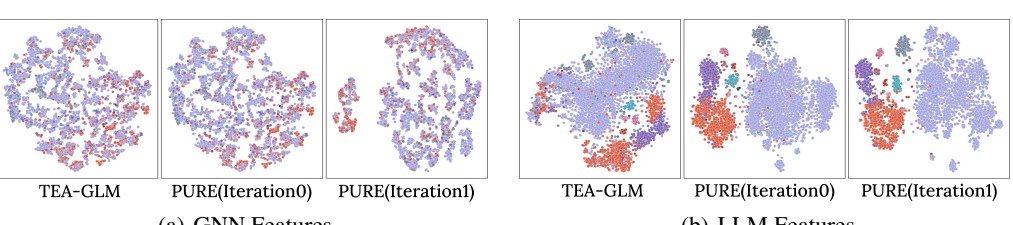

|  | TEA-GLM | PURE(Iteration0) | PURE(Iteration1) | | TEA-GLM | PURE(Iteration0) | PURE(Iteration1) |
|---|---|---|---|---|---|---|---|

(a) GNN Features       (b) LLM Features

Figure 2: **t-SNE visualization** of node embeddings on the History dataset.

For clarity, let $Y_p$ denote the promoted node feature matrix derived from a single prompt instruction, and $Y_p^M$ represent the matrix obtained using mixed prompt instructions. The true node feature matrices are denoted as:

$$X_p = (X, p) \quad \text{and} \quad X_p^M = (X, P),$$

where $p$ is one of $K$ prompt instructions in the set $P$, and $P_{-1}$ is the set of prompt instructions excluding $p$. We assume the true promoted node feature matrix $Y_p^M$ is linearly related to $X_p^M$:

$$Y_p^M = X_p^M W = X_p W_1 + P_{-1} W_2,$$

where $W$ is the weight matrix learned by the neural network, and $W_1$ and $W_2$ are sub-matrices of $W$. This linear relationship reflects that the mixed prompt instructions are computed as the inner product of routed prompts and probability values, i.e., $w^\top P$, and a linear projector $f_{\text{Linear}}^{L_2}(\cdot)$.

The following theorem highlights the potential bias introduced when using a single prompt instruction.

**Theorem 4.1.** *Under the MSE loss, using a single prompt instruction introduces bias in the predicted promoted node feature matrix $Y_p$ relative to the true promoted matrix $Y_p^M$:*

$$Y_p^M - Y_p = (I - X_p(X_p^\top X_p)^{-1} X_p^\top) P_{-1} W_2,$$

*where $I - X_p(X_p^\top X_p)^{-1} X_p^\top$ is a projection matrix.*

The proof of our Theorem 4.1 can be found in Appendix B. Theorem 4.1 shows that without mixed prompting, the bias in $Y_p$ arises from the projection matrix and the weight matrix $W_2$. This bias occurs because a single prompt instruction fails to capture the full complexity of the task. By leveraging diverse mixed prompt strategies, this bias can be significantly reduced, enabling the model to more accurately approximate the true promoted node feature matrix.

## 5 EXPERIMENTS

We conduct experiments on eight widely-used datasets spanning two distinct domains: citation networks (Arxiv (Hu et al., 2020), Pubmed (He et al., 2023), and extended Cora (Wen & Fang, 2023))

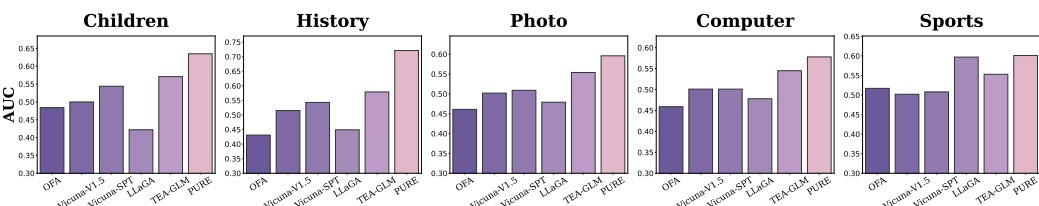

Figure 3: **Cross-task link prediction AUC** across e-commerce datasets.

and e-commerce graphs from the TAG benchmark (Yan et al., 2023) (Children, History, Computer, Photo, and Sports). Each dataset is split into training and test sets following the methodology outlined in TEA-GLM (Wang et al., 2024a). We compare PURE with several baselines, including traditional non-graph neural network approaches (MLP), supervised graph neural network methods (GCN (Kipf & Welling, 2016), GraphSAGE (Hamilton et al., 2017), GAT (Veličković et al., 2018a)), self-supervised methods (DGI (Veličković et al., 2018b)), graph knowledge distillation frameworks (GKD (Yang et al., 2022), GLNN (Zhang et al., 2021)), graph transformer networks (NodeFormer (Wu et al., 2022), DIFFormer (Wu et al., 2023)), large language models (Vicuna-7B-v1.5 (Zheng et al., 2023)), and state-of-the-art models with transfer and zero-shot capabilities (OFA (Liu et al., 2024), GraphGPT (Tang et al., 2024a), LLaGA (Chen et al., 2024a), TEA-GLM (Wang et al., 2024a)). Additional details are provided in Appendix D.

## 5.1 CROSS-DATASET ZERO-SHOT PERFORMANCE

**Setting.** We train the model on the Arxiv dataset (citation domain) and the Computer dataset (e-commerce domain) for node classification, and directly test performance on other datasets within the same domain without any fine-tuning. For GNN-based methods, we preserve the pretrained backbone from the source dataset and only retrain the classifier on the target dataset for prediction. For GraphGPT (Tang et al., 2024a), we report the results of citation datasets provided by the paper. Table 1 shows the zero-shot accuracy on citation and e-commerce datasets.

**Performance Comparison.** From the results, we have three observations. *Firstly*, PURE consistently outperforms competing GNN and LLM predictors across multiple datasets, demonstrating its strong transferability without additional training. *Secondly*, GNN-based methods struggle in zero-shot settings due to their dependence on source graph structures, limiting generalization to domains with different graph properties. While OFA performs well on citation datasets, its performance on e-commerce datasets is hindered by the diverse nature of product types, which challenges its adaptability. In contrast, LLMs excel by leveraging rich pretrained semantic knowledge, enabling them to handle unseen domains better. *Finally*, PURE outperforms both Vicuna-v1.5 and Vicuna-SPT due to its two innovative strategies: (1) Iterative Mutual Alignment Tuning, which facilitates iterative optimization through mutual alignment between GNNs and LLMs, and (2) Mixture of Prompt Expert, which leverages specialized LLM prompts to enhance the performance of GNNs.

**Visualization.** To better understand the learned representations, we visualize the node embeddings of GNN and LLM features for the History dataset using t-SNE, as shown in Figure 2. Since the GNN is not trained with a classification loss, its role is primarily to capture structural information, resulting in a relatively uniform feature distribution. In the initial iteration, the GNN features closely resemble those of TEA-GLM, indicating minimal differentiation. However, after iterative mutual alignment tuning, the GNN features exhibit noticeable clustering patterns, suggesting that mutual prompting effectively integrates text-related features into the GNN representations.

## 5.2 CROSS-TASK ZERO-SHOT PERFORMANCE

**Settings.** We test the model's performance on the link prediction task across all e-commerce datasets after training on the Computer dataset for node classification. The Area Under Curve (AUC) is used as the evaluation metric. The experimental results are summarized in Figure 3.

**Performance Comparison.** Our proposed PURE achieves state-of-the-art performance across all domains, demonstrating strong generalization capabilities for cross-task zero-shot learning. The results reveal three key observations: (1) Pure language model variants (Vicuna) and graph-agnostic approaches (OFA) exhibit fundamental limitations, either ignoring graph topology or suffering from

Table 2: **Ablation study of different variants** on all datasets.

| Variants | Pubmed | Cora | Children | History | Photo | Sports |
|---|---|---|---|---|---|---|
| PURE w/o $f_{\text{Linear}}^G(\cdot)$ | $0.711_{\downarrow 0.134}$ | $0.162_{\downarrow 0.011}$ | $0.243_{\downarrow 0.032}$ | $0.373_{\downarrow 0.221}$ | $0.257_{\downarrow 0.263}$ | $0.289_{\downarrow 0.147}$ |
| PURE w/o $\mathcal{L}_{info}$ | $0.807_{\downarrow 0.038}$ | $0.151_{\downarrow 0.022}$ | $0.263_{\downarrow 0.012}$ | $0.497_{\downarrow 0.097}$ | $0.465_{\downarrow 0.055}$ | $0.405_{\downarrow 0.031}$ |
| PURE w/o $\mathcal{L}_{ins}$ | $0.819_{\downarrow 0.026}$ | $0.152_{\downarrow 0.021}$ | $0.268_{\downarrow 0.007}$ | $0.368_{\downarrow 0.226}$ | $0.519_{\downarrow 0.001}$ | $0.382_{\downarrow 0.054}$ |
| PURE w/o Iteration | $0.835_{\downarrow 0.010}$ | $0.163_{\downarrow 0.010}$ | $0.270_{\downarrow 0.005}$ | $0.538_{\downarrow 0.056}$ | $0.508_{\downarrow 0.012}$ | $0.403_{\downarrow 0.033}$ |
| PURE | **0.845** | **0.173** | **0.275** | **0.594** | **0.520** | **0.436** |

negative transfer, highlighting the necessity of unified modality interaction; (2) Existing graph-text alignment methods show limited capacity to model complex cross-modal dependencies, which often fail to capture the intricate interplay between structural and semantic features, particularly in tasks requiring fine-grained reasoning; (3) While baseline methods show inconsistent performance across domains, PURE maintains robust accuracy by dynamically balancing semantic and structural signals. The results validate that explicit modeling of modality interplay, rather than simple alignment or isolated adaptation, drives effective cross-task transfer in graph-language learning.

### 5.3 ABLATION STUDY

To validate the contribution of core components in PURE, we design four variants:(1) *PURE w/o* $f_{Linear}^G(\cdot)$: Removes the linear projection layer between GNN and LLM, directly feeding raw GNN outputs to LLM; (2) *PURE w/o* $\mathcal{L}_{info}$: Disables feature-wise graph pre-training loss for prototypical prompt learning; (3) *PURE w/o* $\mathcal{L}_{ins}$: Disables instance loss; (4) *PURE w/o Iteration*: Performs single-step inference without multi-round iteration. The results are summarized in Table 2. From the results, we observe four key observations as follows: (1) removing $f_{\text{Linear}}^G(\cdot)$ causes the largest drop, confirming the necessity of embedding-space alignment for effective GNN–LLM interaction; (2) disabling $\mathcal{L}_{info}$ severely degrades semantic-rich tasks (e.g., *History*), indicating its role in capturing fine-grained semantics; (3) the full model surpasses the single-pass variant by 2.1–5.6%, demonstrating the benefit of iterative refinement; (4) removing $\mathcal{L}_{ins}$ yields unstable performance, highlighting its importance in balancing structural and semantic signals.

### 5.4 PARAMETER SENSITIVITY

In this part, we investigate the impact of the number of tokens and the number of iterations. We first vary the number of tokens from 1 to 5 with the other parameter fixed. As shown in Figure 4(a), PURE achieves the best performance with 3 tokens on most datasets, indicating that a moderate number of tokens is crucial for effective mutual alignment tuning. Then

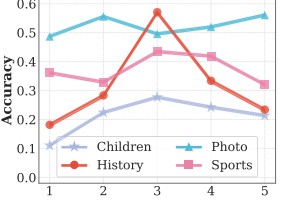
(a) Number of Tokens

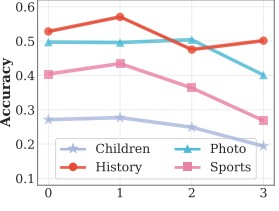
(b) Number of Iterations

Figure 4: Sensitivity analysis to parameters.

the impact of the number of iterations is explored by varying the number of iterations from 0 to 3. Figure 4(b) shows that PURE achieves the best performance with 1 iteration, suggesting that multiple iterations are necessary for effective mutual alignment tuning. However, when the number of iterations increases, the performance decreases. A potential reason is that too many iterations could lead to error accumulation.

## 6 CONCLUSION

In this paper, we propose PURE, a novel framework for zero-shot text-attributed graph learning that synergizes GNNs and LLMs through dual graph pre-training and mutual prompt learning. Dual pre-training captures both instance-level structural relationships and informativeness-aware semantic alignment with LLM token embeddings, enabling transferable graph representations. Mutual prompt learning framework enables iterative enhancement: GNN-derived geometric tokens guide LLM via prototypical projections, while LLM-generated instructions boost GNN performance via prompt experts. Extensive experiments demonstrate PURE's superior performance in cross-dataset and cross-task zero-shot scenarios, achieving state-of-the-art results in node classification and link prediction.

## REPRODUCIBILITY STATEMENT

We have made efforts to ensure the reproducibility of our work. The main components of the proposed PURE framework, including the dual graph pre-training strategy and prototypical prompt learning, are fully described in Sections 4.2–4.3 with all mathematical formulations provided. Hyperparameters, model configurations, dataset details, and the complete training procedure are documented in Section 5 and Appendices D–G. An anonymous link to our implementation is provided at the end of Section 1 to facilitate independent verification.

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

## A LARGE LANGUAGE MODEL (LLM) USAGE STATEMENT

We use the LLM as a general-purpose assistant tool. Specifically, the LLM assists in (i) checking grammar and improving clarity of text descriptions, and (ii) suggesting alternative phrasings for some sections. No parts of the paper are generated entirely by the LLM. All research ideas, experiments, model designs, and results are conceived, implemented, and analyzed solely by the authors. The LLM does not contribute to the development of the methodology, experiments, or analysis presented in this paper. We confirm that the use of the LLM is limited to minor writing support and does not constitute a substantive contribution that would qualify it as a co-author.

## B PROOF OF THEOREM 4.1

Using the mean squared error (MSE) loss, the estimated parameter $\hat{\boldsymbol{W}}$ is obtained by minimizing:

$$\mathcal{L} = \|\boldsymbol{Y}_p^M - \boldsymbol{X}_p \boldsymbol{W}\|^2,$$

which leads to the optimal solution:

$$\hat{\boldsymbol{W}} = (\boldsymbol{X}_p^\top \boldsymbol{X}_p)^{-1} \boldsymbol{X}_p^\top \boldsymbol{Y}_p^M.$$

Substituting this into the expression for the predicted promoted node feature matrix $\boldsymbol{Y}_p$, we obtain:

$$\begin{aligned}
\boldsymbol{Y}_p &= \boldsymbol{X}_p \hat{\boldsymbol{W}} \\
&= \boldsymbol{X}_p (\boldsymbol{X}_p^\top \boldsymbol{X}_p)^{-1} \boldsymbol{X}_p^\top \boldsymbol{Y}_p^M \\
&= \boldsymbol{X}_p (\boldsymbol{X}_p^\top \boldsymbol{X}_p)^{-1} \boldsymbol{X}_p^\top (\boldsymbol{X}_p \boldsymbol{W}_1 + \boldsymbol{P}_{-1} \boldsymbol{W}_2) \\
&= \boldsymbol{X}_p \boldsymbol{W}_1 + \boldsymbol{X}_p (\boldsymbol{X}_p^\top \boldsymbol{X}_p)^{-1} \boldsymbol{X}_p^\top \boldsymbol{P}_{-1} \boldsymbol{W}_2.
\end{aligned} \tag{16}$$

From Equation (16), the difference between the predicted and true promoted node feature matrices is:

$$\begin{aligned}
\boldsymbol{Y}_p - \boldsymbol{Y}_p^M &= \boldsymbol{X}_p \boldsymbol{W}_1 + \boldsymbol{X}_p (\boldsymbol{X}_p^\top \boldsymbol{X}_p)^{-1} \boldsymbol{X}_p^\top \boldsymbol{P}_{-1} \boldsymbol{W}_2 - (\boldsymbol{X}_p \boldsymbol{W}_1 + \boldsymbol{P}_{-1} \boldsymbol{W}_2) \\
&= \boldsymbol{X}_p (\boldsymbol{X}_p^\top \boldsymbol{X}_p)^{-1} \boldsymbol{X}_p^\top \boldsymbol{P}_{-1} \boldsymbol{W}_2 - \boldsymbol{P}_{-1} \boldsymbol{W}_2 \\
&= (\boldsymbol{I} - \boldsymbol{X}_p (\boldsymbol{X}_p^\top \boldsymbol{X}_p)^{-1} \boldsymbol{X}_p^\top) \boldsymbol{P}_{-1} \boldsymbol{W}_2.
\end{aligned} \tag{17}$$

Thus, we conclude the proof of Theorem 4.1.

## C IMPACT STATEMENT

The proposed `PURE` framework advances zero-shot text-attributed graph learning by combining the strengths of GNNs and LLMs. By leveraging dual graph pre-training and mutual prompting, `PURE` enhances the extraction of both structural and semantic information, enabling effective generalization across unseen graphs and tasks. This work has potential applications in domains requiring robust graph analysis, such as social networks, recommender systems, knowledge graphs, and biomedical networks. By reducing the need for task-specific fine-tuning, `PURE` contributes to more efficient and scalable graph-based machine learning.

## D DETAIL OF BASELINES

We compare `PURE` with 14 baseline methods across six technical categories:

**Traditional Non-GNN**: Multi-Layer Perceptron baseline without graph structural awareness, serving as a fundamental reference for non-relational learning.

**Supervised GNNs**:

- GCN (Kipf & Welling, 2016): Spectral graph convolution operator with layer-wise neighborhood aggregation through low-pass frequency filtering.

Table 3: Dataset statistics.

| Domain | Dataset | #Nodes | #Edges | #Classes |
|--------|---------|--------|--------|----------|
| Citation | Arxiv | 169,343 | 1,166,243 | 40 |
| | Pubmed | 19,717 | 44,338 | 3 |
| | Cora | 25,120 | 91,140 | 70 |
| E-commerce | Ele-Computer | 87,229 | 721,081 | 10 |
| | Ele-Photo | 48,362 | 500,928 | 12 |
| | Book-Children | 76,875 | 1,554,578 | 24 |
| | Book-History | 41,551 | 358,574 | 12 |
| | Sports-Fitness | 173,055 | 1,773,500 | 13 |

- GraphSAGE (Hamilton et al., 2017): Inductive framework employing stochastic neighborhood sampling and parameterized aggregation functions.
- GAT (Veličković et al., 2018a): Attention-based architecture with learnable edge importance weights via multi-head attention mechanisms.

**Self-supervised Learning**:

- DGI (Veličković et al., 2018b): Contrastive learning paradigm maximizing mutual information between local node representations and global graph summaries.

**Knowledge Distillation**:

- GKD (Yang et al., 2022): Graph-to-graph distillation framework transferring topological knowledge via adaptive structure matching.
- GLNN (Zhang et al., 2021): Structure-agnostic neural network trained with GNN-generated soft labels for graphless inference.

**Graph Transformers**:

- NodeFormer (Wu et al., 2022): Kernelized transformer architecture enabling efficient all-pair message passing with random feature approximation.
- DIFFormer (Wu et al., 2023): Spectral diffusion-enhanced transformer with adaptive propagation based on eigenbasis decomposition.

**Large Language Models**:

- Vicuna-7B-v1.5 (Zheng et al., 2023): Instruction-following LLM with 7 billion parameters fine-tuned from LLama2.

**State-of-the-Art Models**:

- OFA (Liu et al., 2024): Unified graph foundation model with cross-domain text-graph unification and in-context learning via prompt substructures.
- GraphGPT (Tang et al., 2024a): Graph-text alignment framework with dual-stage instruction tuning and structural-aware projection modules.
- LLaGA (Chen et al., 2024a): Language-graph assistant with topology-preserving sequence reorganization and parameter-efficient graph token projection.
- TEA-GLM (Wang et al., 2024a): GNN-LLM alignment method featuring pretrained representation mapping and unified instruction templates for cross-task generalization.

# E DETAIL OF DATASETS

Table 3 summarizes the key statistics of our evaluation datasets. Below we provide detailed descriptions:

**Citation Networks** focus on academic paper analysis. The **Arxiv** (Hu et al., 2020) dataset contains 169,343 computer science papers from arXiv, where nodes represent publications connected by citations, and labels correspond to 40 subfields. **Pubmed** (He et al., 2023) includes 19,717 diabetes-related papers categorized into three clinical types (Type 1/2 Diabetes and Experimentally Induced Diabetes), with edges reflecting citation relationships. The extended **Cora** (Wen & Fang, 2023) dataset expands the classic version to 25,120 machine learning papers and 70 fine-grained research topics, capturing broader taxonomy.

**E-commerce Datasets** from the TAG benchmark model product relationships (Yan et al., 2023). **Book-Children** (76,875 nodes) and **Book-History** (41,551 nodes) represent Amazon book subcategories with three-level hierarchical labels. **Ele-Computer** (87,229 nodes) and **Ele-Photo** (48,362 nodes) cover electronics products with functional categorizations. **Sports-Fitness** (173,055 nodes) is the largest dataset, where edges encode co-purchasing patterns between fitness-related items. All e-commerce edges are derived from co-viewing or co-buying behaviors, with labels reflecting product taxonomies.

## F  IMPLEMENTATION DETAILS

The framework operates in two phases. During **Dual Graph Pre-training**, we initialize a 2-layer GraphSAGE backbone with mean aggregation and ReLU activation, setting the hidden dimension to 4,096 to align with Vicuna-7B's token embeddings. This phase uses AdamW optimization with a batch size of 512 for 60 epochs, learning structural patterns from raw graph data.

In the **Prototypical Prompt Learning** phase, we reduce the batch size to 2 to accommodate memory constraints when integrating the LLM. The alignment process employs three trainable tokens to bridge GNN and LLM representations, optimized with a learning rate of $1 \times 10^{-3}$ for one full iteration over the dataset. Experiments run on four NVIDIA A100 GPUs (80GB memory) with an 80-10-10 data split following TEA-GLM's protocol. For evaluation, we report accuracy and macro-F1 for node classification, and AUC-ROC for link prediction, ensuring consistency with graph learning benchmarks.

For loss computation, the total loss in `PURE` combines step-wise components for each direction of the iterative mutual prompting process. In the forward step (GNN $\rightarrow$ LLM), we train a linear projector using an MSE loss to align the GNN embedding space to the LLM token space. In the backward step (LLM $\rightarrow$ GNN), we optimize

$$\mathcal{L}_{\text{total}} = \mathcal{L} + \lambda \left( \mathcal{L}_{ind} + \mathcal{L}_{imp} \right), \tag{18}$$

where $\mathcal{L} = \frac{1}{2}(\mathcal{L}_{ins} + \mathcal{L}_{info})$ is the contrastive loss, and $\mathcal{L}_{ind}$ and $\mathcal{L}_{imp}$ are regularization terms to mitigate collapse. We set $\lambda = 0.2$ to control the weights of $\mathcal{L}_{ind}$ and $\mathcal{L}_{imp}$ in the total loss.

## G  PSEUDOCODE OF `PURE`

This section presents the training flow of `PURE` as pseudocode with equation references.

---

**Algorithm 1:** Dual Graph Pre-training

---

**Inputs:** graph $\mathcal{G} = (\boldsymbol{A}, \boldsymbol{X})$;
**Hyperparameters:** $T$, $p_r$, $p_m$, $\tau$; PCA over LLM tokens;
**for** $t = 1, \cdots, T$ **do**

    Sample $\widetilde{\boldsymbol{R}} \sim \mathcal{B}(1 - p_r)$, set $\widetilde{\boldsymbol{A}} = \boldsymbol{A} \circ \widetilde{\boldsymbol{R}}$ ;         // Eq. 2

    Sample $\widetilde{\boldsymbol{m}} \sim \mathcal{B}(1 - p_m)$, form masked $\widetilde{\boldsymbol{X}}$ ;         // Eq. 3

    $\boldsymbol{Z}^* = f_{\text{GNN}}(\widetilde{\boldsymbol{A}}^*, \widetilde{\boldsymbol{X}}^*)$ ;         // Eq. 4

    Compute $\mathcal{L}_{ins}$ using $\ell(\cdot)$ and $\phi(\cdot)$ ;         // Eqs. 5-- 6

    PCA over LLM tokens $\Rightarrow$ top comps $\boldsymbol{C}$; $\boldsymbol{Z}^* \leftarrow \boldsymbol{Z}^* \boldsymbol{C}^{\text{T}}$ ;         // Eq. 7

    Compute informativeness loss $\mathcal{L}_{info}$ ;         // Eq. 8

    Update $f_{\text{GNN}}$ by $\mathcal{L} = \frac{1}{2}(\mathcal{L}_{ins} + \mathcal{L}_{info})$ ;         // Eq. 9

**Return:** pre-trained encoder $f_{\text{GNN}}$;

---

---

**Algorithm 2:** Prototypical Prompt Learning with Mutual Alignment

---

**Inputs:** $f_{\text{GNN}}$, frozen LLM, prototypes $K$, iterations $I$;
// Initial GNN processing
$\boldsymbol{Z} = f_{\text{GNN}}(\boldsymbol{A}, \boldsymbol{X})$ ;                               // Get initial node embeddings
// Initial LLM processing
For node $v$ with $\boldsymbol{z}_v$, compute $\boldsymbol{h}_v^k = f_{\text{Linear}}^{G,k}(\boldsymbol{z}_v)$, $k = 1..K$ ;                               // Eq. 10
Form $\boldsymbol{H}_v = \{\boldsymbol{h}_v^k\}_{k=1}^K$ and use as soft tokens in LLM instruction;
**for** $i = 1, \cdots, I$ **do**
    // LLM to GNN: Language prompting for GNN
    From LLM outputs $\{\boldsymbol{h}_k\}_{k=1}^K$, set $\boldsymbol{p}^k = f_{\text{Linear}}^{L1,k}(\boldsymbol{h}_k)$ ;                               // Eq. 11
    $w_k(\boldsymbol{z}_v) = [\text{Softmax}(f_{\text{Linear}}^R(\boldsymbol{z}_v) \circ (1 + \delta))]_k$ ;                               // Eq. 12
    $\boldsymbol{x}_v' = [\boldsymbol{x}_v; \boldsymbol{w}^{\text{T}}\boldsymbol{P}]$, $\boldsymbol{x}_{v,p} = f_{\text{Linear}}^{L2}(\boldsymbol{x}_v')$ ;                               // Eq. 13
    Freeze LLM; train $\{f_{\text{Linear}}^{L1,k}\}$, $f_{\text{Linear}}^{L2}$, $f_{\text{Linear}}^R$ with $\mathcal{L}$, $\mathcal{L}_{ind}$, and $\mathcal{L}_{imp}$ ;                               // Eq. 9, 15
    // GNN to LLM: Geometric prompting for LLM
    Update $\boldsymbol{Z} = f_{\text{GNN}}(\boldsymbol{A}, \boldsymbol{X}_p)$ with prompted features;
    For node $v$ with updated $\boldsymbol{z}_v$, compute $\boldsymbol{h}_v^k = f_{\text{Linear}}^{G,k}(\boldsymbol{z}_v)$, $k = 1..K$ ;                               // Eq. 10
    Form $\boldsymbol{H}_v = \{\boldsymbol{h}_v^k\}_{k=1}^K$ and add $\mathcal{L}_{ind}$ ;                               // Eq. 14
    Use $\boldsymbol{H}_v$ as soft tokens in LLM instruction; update $\{f_{\text{Linear}}^{G,k}\}$ on task loss $+\mathcal{L}_{ind}$;
// Final prediction with LLM
Generate final predictions using LLM with aligned prompts;
**Return:** projectors $\{f_{\text{Linear}}^{G,k}\}$, $\{f_{\text{Linear}}^{L1,k}\}$, $f_{\text{Linear}}^{L2}$, router $f_{\text{Linear}}^R$;

---

## H  COMPLEXITY ANALYSIS

We analyze the computational complexity of the proposed `PURE` framework by breaking it down into its major components.

**PCA Pre-computation.**  The Principal Component Analysis (PCA) on the LLM token embeddings is a one-time offline operation performed before training. This pre-computation cost is negligible during model training and inference.

**Graph Model Pre-training.**  For each forward pass of the GNN encoder, the time complexity is $O(|E|d)$, where $|E|$ denotes the number of edges in the graph and $d$ is the hidden dimension. This phase is executed once to obtain transferable node representations.

**Mutual Prompting.**  During the mutual prompting stage, the GNN outputs are projected through linear layers with a complexity of $O(d^2)$. On the LLM side, linear projections and prompt routing with $K$ experts introduce an additional complexity of $O(Kd^2)$. As the mutual prompting process is iterated $I$ times, the total time complexity of the alignment process can be expressed as:

$$O\big((I + 1)(|E|d + d^2) + IKd^2\big).$$

Overall, the cost of PCA pre-computation and graph pre-training is incurred once, while the mutual prompting cost scales with the number of iterations $I$ and the number of prompt experts $K$. This analysis shows that `PURE` maintains linear complexity with respect to the number of edges and quadratic complexity with respect to the hidden dimension $d$, which is practical for large-scale text-attributed graphs.

## I  MORE EXPERIMENTAL RESULTS

### I.1  LEGITIMACY EXPERIMENT

To assess the model's capability of generating valid responses under open-ended scenarios, we conduct legality evaluation following the methodology in (Zhang et al., 2024). This experiment measures the model's ability to produce answers strictly conforming to predefined formats and

Table 4: Legality rate of LLM-backbone model

| Model | Seen | | Unseen | | | | | |
|---|---|---|---|---|---|---|---|---|
| | Arxiv | Computer | Pubmed | Cora | Children | History | Photo | Sports |
| Vicuna-7B-v1.5 | 99.3 | 96.7 | 100.0 | 95.8 | 99.2 | 98.9 | 94.1 | 99.6 |
| LLaGA | 100.0 | 100.0 | 98.9 | 79.9 | 93.1 | 92.4 | 77.8 | 94.3 |
| TEA-GLM | 100.0 | 100.0 | 100.0 | 92.6 | 97.0 | 99.6 | 99.2 | 98.5 |
| PURE | 100.0 | 100.0 | 100.0 | 75.7 | 98.2 | 99.6 | 99.5 | 99.1 |

Table 5: Macro F1 of node classification task (**bold** highlights the best result across all methods, while underline highlights the second-best results).

| Model | Pubmed | Cora | Children | History | Photo | Sports |
|---|---|---|---|---|---|---|
| MLP | $0.246 \pm 0.042$ | $0.009 \pm 0.004$ | $0.007 \pm 0.007$ | $0.023 \pm 0.008$ | $0.041 \pm 0.023$ | $0.019 \pm 0.005$ |
| *GNN as Predictor* | | | | | | |
| GCN | $0.187 \pm 0.021$ | $0.007 \pm 0.001$ | $0.006 \pm 0.004$ | $0.024 \pm 0.013$ | $0.034 \pm 0.007$ | $0.017 \pm 0.009$ |
| GraphSAGE | $0.257 \pm 0.084$ | $0.007 \pm 0.003$ | $0.005 \pm 0.003$ | $0.029 \pm 0.024$ | $0.020 \pm 0.011$ | $0.021 \pm 0.004$ |
| GAT | $0.259 \pm 0.065$ | $0.006 \pm 0.001$ | $0.063 \pm 0.067$ | $0.159 \pm 0.117$ | $0.036 \pm 0.035$ | $0.091 \pm 0.090$ |
| DGI | $0.213 \pm 0.127$ | $0.004 \pm 0.002$ | $0.012 \pm 0.004$ | $0.038 \pm 0.015$ | $0.045 \pm 0.015$ | $0.018 \pm 0.005$ |
| GKD | $0.247 \pm 0.039$ | $0.004 \pm 0.001$ | $0.028 \pm 0.003$ | $0.060 \pm 0.008$ | $0.049 \pm 0.015$ | $0.050 \pm 0.008$ |
| GLNN | $0.221 \pm 0.033$ | $0.006 \pm 0.001$ | $0.021 \pm 0.003$ | $0.064 \pm 0.007$ | $0.057 \pm 0.002$ | $0.052 \pm 0.003$ |
| NodeFormer | $0.232 \pm 0.089$ | $0.008 \pm 0.003$ | $0.019 \pm 0.008$ | $0.046 \pm 0.031$ | $0.055 \pm 0.006$ | $0.049 \pm 0.009$ |
| DIFFormer | $0.187 \pm 0.007$ | $0.007 \pm 0.002$ | $0.002 \pm 0.002$ | $0.050 \pm 0.019$ | $0.069 \pm 0.010$ | $0.045 \pm 0.007$ |
| OFA | $0.287 \pm 0.059$ | $0.091 \pm 0.013$ | $0.017 \pm 0.010$ | $0.026 \pm 0.007$ | $0.103 \pm 0.007$ | $0.043 \pm 0.021$ |
| *LLM as Predictor* | | | | | | |
| Vicuna-7B-v1.5 | $0.629 \pm 0.024$ | $0.109 \pm 0.002$ | $\mathbf{0.279 \pm 0.002}$ | $0.349 \pm 0.003$ | $0.383 \pm 0.001$ | $0.410 \pm 0.002$ |
| GraphGPT-std | 0.649 | 0.082 | - | - | - | - |
| GraphGPT-cot | 0.482 | 0.127 | - | - | - | - |
| LLaGA | $0.778 \pm 0.056$ | $0.108 \pm 0.014$ | $0.163 \pm 0.029$ | $0.144 \pm 0.025$ | $0.362 \pm 0.039$ | $\underline{0.446 \pm 0.035}$ |
| TEA-GLM | $\underline{0.839 \pm 0.012}$ | $0.148 \pm 0.015$ | $0.252 \pm 0.005$ | $\underline{0.365 \pm 0.011}$ | $\mathbf{0.421 \pm 0.032}$ | $0.430 \pm 0.009$ |
| PURE | $\mathbf{0.841 \pm 0.010}$ | $\mathbf{0.165 \pm 0.017}$ | $\underline{0.264 \pm 0.004}$ | $\mathbf{0.374 \pm 0.012}$ | $\underline{0.417 \pm 0.008}$ | $\mathbf{0.457 \pm 0.013}$ |

semantic constraints, particularly when handling unseen domains. The legality rate is calculated as the proportion of responses that satisfy content constraints (e.g., valid label candidates).

As shown in Table 4, our preliminary results on seen datasets (Arxiv and Computer domains) demonstrate that PURE achieves perfect legality rates (100%), indicating strong alignment with format specifications through our instruction tuning strategy. For unseen domains, observations suggest our model maintains stable text generation compared to baseline methods. This can be attributed to our hybrid training approach that combines semantic understanding with structural constraints, effectively reducing errors in unfamiliar scenarios.

### I.2 F1 SCORE ON NODE CLASSIFICATION TASK

F1 score on the node classification task is shown in Table 5.

### I.3 SUPERVISED RESULTS

Table 6 shows the accuracy and macro F1 on training datasets. Due to the lack of supervised loss during the GNN pre-training phase, PURE does not achieve the best results on seen domains. However, it still outperforms most baseline methods, demonstrating the effectiveness of our approach.

### I.4 SCALABILITY OF PURE TO DIFFERENT LLM SIZES

While our main experiments employ Vicuna-7B (Zheng et al., 2023) as the language model backbone, the proposed PURE framework is model-agnostic and can be readily applied to smaller LLMs. To evaluate the impact of model size, we replace Vicuna-7B with LLaMA-3.2-3B[1] and conduct zero-shot node classification on four benchmark datasets. The results are reported in Table 7.

---

[1]https://huggingface.co/meta-llama/Llama-3.2-3B-Instruct

Table 6: Accuracy and macro F1 on training datasets (**bold** highlights the best result across all methods, while underline highlights the second-best results).

| Model | Arxiv | | Computer | |
|---|---|---|---|---|
| | Acc | F1 | Acc | F1 |
| MLP | $0.546 \pm 0.004$ | $0.295 \pm 0.007$ | $0.420 \pm 0.006$ | $0.267 \pm 0.005$ |
| *GNN as Predictor* | | | | |
| GCN | $0.545 \pm 0.005$ | $0.317 \pm 0.006$ | $0.424 \pm 0.012$ | $0.386 \pm 0.014$ |
| GraphSAGE | $0.556 \pm 0.006$ | $0.315 \pm 0.008$ | $0.534 \pm 0.037$ | $0.347 \pm 0.036$ |
| GAT | $0.561 \pm 0.003$ | $0.339 \pm 0.005$ | $0.609 \pm 0.035$ | $0.598 \pm 0.039$ |
| DGI | $0.342 \pm 0.024$ | $0.336 \pm 0.011$ | $0.594 \pm 0.004$ | $0.452 \pm 0.008$ |
| GKD | $0.393 \pm 0.085$ | $0.164 \pm 0.029$ | $0.351 \pm 0.031$ | $0.155 \pm 0.016$ |
| GLNN | $0.602 \pm 0.004$ | $0.362 \pm 0.008$ | $0.393 \pm 0.005$ | $0.243 \pm 0.007$ |
| NodeFormer | $0.544 \pm 0.016$ | $0.297 \pm 0.029$ | $0.434 \pm 0.012$ | $0.288 \pm 0.012$ |
| DIFFormer | $0.616 \pm 0.025$ | $0.356 \pm 0.024$ | $0.629 \pm 0.012$ | $0.467 \pm 0.022$ |
| OFA | $\underline{0.682 \pm 0.006}$ | $\underline{0.495 \pm 0.006}$ | $\mathbf{0.753 \pm 0.004}$ | $\mathbf{0.687 \pm 0.006}$ |
| *LLM as Predictor* | | | | |
| Vicuna-7B-v1.5 | $0.347 \pm 0.000$ | $0.164 \pm 0.001$ | $0.372 \pm 0.010$ | $0.304 \pm 0.002$ |
| GraphGPT-std | 0.626 | 0.262 | - | - |
| GraphGPT-cot | 0.576 | 0.228 | - | - |
| LLaGA | $\mathbf{0.749 \pm 0.001}$ | $\mathbf{0.575 \pm 0.003}$ | $\underline{0.642 \pm 0.004}$ | $\underline{0.562 \pm 0.001}$ |
| TEA-GLM | $0.655 \pm 0.001$ | $0.445 \pm 0.002$ | $0.578 \pm 0.002$ | $0.496 \pm 0.010$ |
| PURE | $0.631 \pm 0.008$ | $0.412 \pm 0.007$ | $0.580 \pm 0.002$ | $0.510 \pm 0.008$ |

Table 7: Zero-shot node classification performance of PURE across different language model sizes.

| Model | Children | History | Photo | Sports |
|---|---|---|---|---|
| LLaMA-3B | 0.267 | 0.350 | 0.448 | 0.328 |
| Vicuna-7B-v1.5 | 0.270 | 0.363 | 0.378 | 0.370 |
| TEA-GLM | 0.271 | 0.528 | 0.497 | 0.404 |
| PURE (LLaMA-3B) | 0.274 | 0.318 | 0.501 | 0.396 |
| **PURE (Vicuna-7B)** | **0.275** | **0.594** | **0.520** | **0.436** |

As shown in Table 7, although the base performance of the smaller LLaMA-3B model is lower than that of Vicuna-7B on several datasets, applying PURE consistently yields performance gains. This demonstrates that PURE maintains robust transferability and scalability across LLM sizes, enhancing zero-shot graph learning even with smaller parameter LLMs.

