# OpenReview forum: "PURE: Prototypical Mutual Prompting Enhancement for Zero-shot Text-attributed Graph Learning"
_ICLR.cc/2026/Conference — ICLR 2026 Conference Withdrawn Submission_

### Official Review · Reviewer_w2J1 · 2025-10-31

**Soundness:** 2
**Presentation:** 3
**Contribution:** 3
**Rating:** 6
**Confidence:** 3

**Summary:**

This paper addresses the challenge of zero-shot text-attributed graph (TAG) learning—a task where models must generate high-quality node representations for unseen TAGs (no labeled data from target domains/tasks). Existing methods often prioritize either large language models (LLMs) (strong semantic generalization but poor structural encoding) or graph neural networks (GNNs) (effective structure capture but limited transferability). To bridge this gap, the authors propose PURE (Prototypical Mutual Prompting Enhancement), a framework that fuses GNNs and LLMs via dual pre-training and iterative mutual prompting.

**Strengths:**

1.PURE addresses a critical limitation of prior work: the one-sided integration of GNNs and LLMs (e.g., LLM-as-Enhancer for GNNs, GNN-as-Structural-Encoder for LLMs). By designing mutual prompting, the framework creates a closed loop where structural and semantic information iteratively refine each other—this is a creative combination of prompt learning (traditionally used for NLP/CV) and graph learning.

2.The dual pre-training (instance + informativeness views) solves a longstanding issue: GNNs pre-trained on structure alone often fail to align with LLM semantic spaces, while LLMs lack structural awareness. The use of prototypical projections (K expert prompts) further distinguishes PURE from single-prompt methods, as it explicitly models diverse task semantics (avoiding "prompt collapse").

3.The authors test PURE on 8 datasets across two domains (citation networks: Arxiv, Pubmed, Cora; e-commerce: Children, History, Photo, etc.) and compare it to a comprehensive set of baselines (traditional MLPs, supervised GNNs, self-supervised methods, LLMs, and SOTA TAG models). They also include ablation-adjacent analyses (e.g., t-SNE visualizations of embeddings before/after mutual prompting) to validate that PURE improves feature clustering.

4.Zero-shot TAG learning is highly relevant to real-world applications where labeled data is scarce (e.g., social network analysis, e-commerce recommendation, biomedical knowledge graphs). PURE’s strengths address practical needs the scalability: By freezing GNN/LLM backbones and only training lightweight projectors, PURE avoids expensive fine-tuning of large models.

**Weaknesses:**

1.The paper focuses exclusively on citation networks (academic papers) and e-commerce graphs (product co-purchasing). While these are valid TAG domains, they lack diversity in terms of:

a. Graph Structure: Citation/e-commerce graphs are mostly homogeneous and sparse, but real-world TAGs often include heterogeneous graphs (e.g., social media graphs with user/post/comment nodes) or dense graphs (e.g., biomedical networks with protein-drug interactions).

b. Text Attribute Complexity: The text attributes here (paper titles, product names) are short and structured, but many TAGs have unstructured text (e.g., user reviews, clinical notes).

2.The paper describes GNN-generated "geometric tokens" and LLM-generated "task instructions" but provides no qualitative examples of these prompts. For instance:

a. What does a GNN-derived token look like (e.g., is it a vector, or a text snippet)?
b. How do LLM instructions differ between node classification and link prediction?
c. Do the prototypical prompts adapt to different domains (e.g., citation vs. e-commerce)?

Qualitative analysis would help researchers understand why mutual prompting works (e.g., whether GNN tokens truly encode structural information that LLMs miss) and not just that it works.

**Questions:**

1.How do you select the number of prototypical experts (K) for a given TAG? Is there an adaptive method to choose K based on graph/text complexity (e.g., more experts for heterogeneous graphs)? Have you observed cases where increasing K beyond a threshold leads to overfitting?

2.Have you tested PURE with LLMs optimized for specific domains (e.g., BioBERT for biomedical TAGs, or BERT4Rec for e-commerce)? Does PURE’s prompting framework adapt to domain-specific LLMs, or does it require retuning of projectors?

---

### Official Review · Reviewer_GnXE · 2025-11-01

**Soundness:** 2
**Presentation:** 3
**Contribution:** 2
**Rating:** 2
**Confidence:** 5

**Summary:**

The paper proposes a Prototypical Mutual Prompting Enhancement framework (PURE) for zero-shot text-attributed graph learning, which integrates large language models and graph neural networks through prototypical prompt learning. By combining dual graph pretraining with mutual prompting between frozen LLM and graph models, PURE effectively captures both semantic and structural information.

**Strengths:**

1. The paper is well-organized and clearly written, making it easy to follow.
2. The proposed method demonstrates strong performance on both cross-dataset zero-shot learning and cross-task link prediction tasks.

**Weaknesses:**

1. I have concerns regarding the originality and contribution of this work. The authors claim to pioneer a new perspective by introducing prompt learning to combine the strengths of language models and graph models for zero-shot text-attributed graph learning. Another contribution is the novel methodology that the proposed PURE framework leverages graph models to extract geometric relationships for language model prompting and employs prototypical projections to generate text-based prompts for enhancing graph representations. However, similar ideas have already been explored in prior work. For example, [1] introduced a multi-modal prompt learning paradigm that jointly learns graph and text prompts to adapt pre-trained GNNs to downstream tasks, and further demonstrated a CLIP-style zero-shot classification prototype that generalizes GNNs to unseen classes under weak text supervision. Given the conceptual overlap between this paper and [1], the claimed novelty and contribution of PURE appear limited, and the work may be better positioned as an incremental extension rather than a pioneering framework.
[1] Zihao Li, Lecheng Zheng, Bowen Jin, Dongqi Fu, Baoyu Jing, Yikun Ban, Jingrui He, and Jiawei Han. 2025. Can Graph Neural Networks Learn Language with Extremely Weak Text Supervision?. In Proceedings of the 63rd Annual Meeting of the Association for Computational Linguistics (Volume 1: Long Papers), pages 11138–11165, Vienna, Austria. Association for Computational Linguistics.

2. In the theoretical analysis, the authors claim that "Theorem 4.1 shows that without mixed prompting, the bias in arises from the projection matrix and the weight matrix. This bias occurs because a single prompt instruction fails to capture the full complexity of the task. By leveraging diverse mixed prompt strategies, this bias can be significantly reduced, enabling the model to more accurately approximate the true promoted node feature matrix." However, Theorem 4.1 only demonstrates that using a single prompt instruction introduces bias and it does not theoretically establish that employing diverse mixed prompt strategies necessarily reduces this bias. So the claimed bias reduction remains an intuitive hypothesis rather than a proven result.

3. In figure 2, if the text prompt can help improve the graph prompt, why are LLM features much better than the GNN features?

4. Based on the results shown in figure 4, the proposed method is very sensitive to the number of tokens. However, in the zero-shot setting, it's almost impossible to determine the best number of tokens.

**Questions:**

1. In figure 2, if the text prompt can help improve the graph prompt, why are LLM features much better than the GNN features?

---

### Official Review · Reviewer_yzpZ · 2025-11-01

**Soundness:** 2
**Presentation:** 3
**Contribution:** 2
**Rating:** 4
**Confidence:** 2

**Summary:**

The paper proposes PURE, a mutual prompting framework that aligns a pre-trained GNN with a large language model for zero-shot learning on text-attributed graphs. It uses dual graph pre-training and a mixture-of-prototype prompts to allow cross-dataset and cross-task generalization.

**Strengths:**

1. The authors leverage both semantic and structural information effectively.
2. PURE shows strong zero-shot performance across datasets and tasks.
3. The paper is well written and easy to follow.

**Weaknesses:**

1. The novelty is limited. The overall framework and idea seems similar to previous LLM+GNN works.
2. Training pipeline is complex with multiple alignment stages.

**Questions:**

See weaknesses.

---

### Official Review · Reviewer_LCrY · 2025-11-01

**Soundness:** 2
**Presentation:** 3
**Contribution:** 3
**Rating:** 6
**Confidence:** 2

**Summary:**

The paper proposes PURE for zero-shot text-attributed graph learning. It first pretrains a GNN with (i) instance-level contrast and (ii) an informativeness-aware alignment that projects node reps onto principal components of LLM token embeddings. Then it performs mutual prompting: GNN-derived “geometric tokens” act as soft prompts to the LLM, while LLM outputs are mapped back as soft prompts for the GNN with gating and anti-collapse regularizers. Across cross-dataset node classification and cross-task link prediction, PURE surpasses GNN/LLM baselines.

**Strengths:**

- Bidirectional integration rather than one-way alignment: GNN→LLM and LLM→GNN prompts improve transfer.
- Prototypical prompt experts reduce prompt bias; backed by a simple theorem analyzing bias from single-prompt training.
- Dual pretraining (instance + token-PC alignment) yields transferable graph reps that better match LLM semantics.

**Weaknesses:**

-   The pipeline stacks multiple objectives (contrastive, informativeness alignment), projections, gating, and iterative alignment. This enlarges the hyperparameter/search space and can introduce training instability.

-  Results hinge on Vicuna-7B embeddings. There’s no backbone/size sweep or comparison to stronger prompt-engineered LLM baselines, so the method’s generality and robustness across LLMs remain unclear.

- Key transformer-based graph learners are not included as baselines for compaision, leaving open how PURE compares against the current state of the art in graph transformers.

**Questions:**

See weakness.

---

### Note · Authors · 2026-01-05

I have read and agree with the venue's withdrawal policy on behalf of myself and my co-authors.